# High-Level Production of scFv-Fc Antibody Using an Artificial Promoter System with Transcriptional Positive Feedback Loop of Transactivator in CHO Cells

**DOI:** 10.3390/cells12222638

**Published:** 2023-11-16

**Authors:** Binbin Ying, Yoshinori Kawabe, Feiyang Zheng, Yuki Amamoto, Masamichi Kamihira

**Affiliations:** Department of Chemical Engineering, Faculty of Engineering, Kyushu University, 744 Motooka, Nishi-ku, Fukuoka 819-0395, Japan; ying.binbin.538@s.kyushu-u.ac.jp (B.Y.); kawabe@chem-eng.kyushu-u.ac.jp (Y.K.); zfylyhzw@gmail.com (F.Z.); amamoto_y@chem-eng.kyushu-u.ac.jp (Y.A.)

**Keywords:** artificial transactivator, transcriptional amplification, CHO cell, scFv-Fc

## Abstract

With the increasing demand for therapeutic antibodies, CHO cells have become the de facto standard as producer host cells for biopharmaceutical production. High production yields are required for antibody production, and developing a high-titer production system is increasingly crucial. This study was established to develop a high-production system using a synthetic biology approach by designing a gene expression system based on an artificial transcription factor that can strongly induce the high expression of target genes in CHO cells. To demonstrate the functionality of this artificial gene expression system and its ability to induce the high expression of target genes in CHO cells, a model antibody (scFv-Fc) was produced using this system. Excellent results were obtained with the plate scale, and when attempting continuous production in semi-continuous cultures using bioreactor tubes with high-cell-density suspension culture using a serum-free medium, high-titer antibody production at the gram-per-liter level was achieved. Shifting the culture temperature to a low temperature of 33 °C achieved scFv-Fc concentrations of up to 5.5 g/L with a specific production rate of 262 pg/(cell∙day). This artificial gene expression system should be a powerful tool for CHO cell engineering aimed at constructing high-yield production systems.

## 1. Introduction

In recent years, the demand for the production of biopharmaceuticals, such as therapeutic antibodies for cancer and rheumatoid arthritis, has rapidly expanded due to their high therapeutic efficacy [1]. More than 70% of therapeutic antibodies are currently produced in cell culture using Chinese hamster ovary (CHO) cells as host production cells [2]. CHO cells can produce bioactive, non-immunogenic recombinant antibodies with glycosylation patterns similar to those of human antibodies [3]. However, the slow growth of cells compared with that of microorganisms and the consumption of large amounts of medium lead to increased production costs [4]. To reduce manufacturing costs, cell line development [5], medium development [6], and culture environment and process optimization [7], among others, have been performed to improve production titers by increasing specific productivity or viable cell density. Several studies have achieved high-titer antibody production at concentrations on the order of grams per liter by improving culture processes such as high-cell-density fed-batch culture and continuous culture [8,9]. Cell line development, which is the most upstream process of antibody manufacturing, is very important because the cell line determines the performance of the entire manufacturing process. Improving the system for expressing the target gene is effective to further elevate the antibody production level.

Various studies have been conducted on the constituent elements of gene expression cassettes to achieve high expression of the target gene, including insulators that are less susceptible to gene silencing [10,11], improved transcription termination at gene insertion sites in the genome [12], stabilization of target transcripts [13], and codon optimization [14]. Because gene expression is triggered by transcription from promoters, the ability to induce strong gene expression has been developed from both natural and synthetic promoters [15]. The tetracycline-dependent transcription activation system is a synthetic promoter that works in animal cells [16]. This system consists of a fusion protein of the tetracycline repressor (TetR), as a DNA-binding domain, the transcriptional activation domain (TAD), as an artificial transcription factor (aTF), and an artificial promoter with a TetR-responsive element (TRE) sequence. The gene expression system can strongly induce the expression of a target gene. We have constructed gene expression control systems that work in animal cells using a synthetic biology approach. In these systems, the overexpression of aTF is induced by transcriptional amplification using a positive feedback loop of the aTF expression in the Tet-transcription activation system, inducing high-level expression of target genes under the control of an artificial promoter that responds to aTF. This system not only regulates target gene expression by adding inducer drugs, which is the original induction method of the Tet-transcription activation system, but also allows for autonomous induction according to external environmental factors by selecting a trigger promoter that induces aTF expression. In fact, we have succeeded in generating cell lines that can express target genes in response to various external environmental factors, such as hypoxia [17] and heat treatment caused by direct heating or heat generation of magnetic nanoparticles upon the application of a magnetic field [18,19].

In this study, we applied an artificial gene expression system with the amplification of aTF expression via a positive feedback loop to CHO cells, and aimed to develop a method for the high production of recombinant antibodies. An anti-prion single-chain antibody fragment fused with a human IgG1-derived Fc region (scFv-Fc) was used as a model antibody. To increase the number of transgenes, a PiggyBac transposon vector system was used to integrate an aTF expression unit with a positive feedback loop and a target gene expression unit into the CHO cell genome. Recombinant protein production via CHO cell culture is improved by culturing at lower temperatures [20,21]. Therefore, the scFv-Fc antibody productivity of our artificial gene expression system in combination with low-temperature culture was evaluated. After the constructed recombinant cells were adapted to serum-free suspension culture, the efficacy of the gene expression system in a serum-free medium, the effect of enhancing production in low-temperature culture, and the structures of glycans modified to scFv-Fc were evaluated. Furthermore, antibody productivity, glucose usage, and the transcriptome in semi-continuous culture with high cell density were analyzed under normal- and low-temperature conditions.

## 2. Materials and Methods

### 2.1. Cells and Media

CHO-K1 (RIKEN Cell Bank, Tsukuba, Japan) and recombinant CHO cells were cultured using F12 medium (Sigma-Aldrich, St. Louis, MO, USA) supplemented with 10% fetal calf serum (FCS; BioWest, Nuaillé, France) and antibiotics (Penicillin-Streptomycin, #15140122; Invitrogen, Waltham, MA, USA). The cells were cultured in a 5% CO_2_ incubator at 37 °C or 33 °C. Serum-free adaptation was performed by gradually reducing the serum concentration using a mixture of 10% FCS-containing F12 medium and serum-free medium. The serum-free medium was prepared as follows: CD CHO AGT Medium (#12490025; Invitrogen), CHO custom medium (#ISJGp014; Fujifilm Wako Pure Chemical, Osaka, Japan), and EX-CELL Advanced CHO Fed-batch Medium (#14366C-1000ML; Sigma-Aldrich) were mixed at a 3:1:1 ratio. The medium was supplemented with 0.2% anti-clumping reagent (#0010057AE; Invitrogen), 4 mM L-alanyl-L-glutamine (#G8541; Sigma-Aldrich), and antibiotics (Penicillin-Streptomycin). When culturing cells in serum-containing medium, 100 mm cell culture dishes (BioLite #130182; Thermo Fisher Scientific, Waltham, MA, USA) or 24-well tissue culture plates (BioLite #130186; Thermo Fisher Scientific) were used. For serum-free suspension culture, the cells were cultured with 10 mL of culture medium in 50 mL bioreactor tubes equipped with sterile gas exchange caps (#87050, TPP; Techno Plastic Products AG, Trasadingen, Switzerland). The tubes were placed at a 45° angle and rotated at a speed of 180 rpm in a shaking incubator (Model #0081704-000; Taitec, Koshigaya, Japan). All cell cultures were conducted within a 5% CO_2_ incubator at 37 °C or 33 °C.

### 2.2. Plasmid Construction

DNA fragments for the TRE promoter consisting of TRE and cytomegalovirus (CMV) minimal promoter, aTF, and the reporter (GFP-2A-Puro) genes were chemically synthesized (GeneArt Gene Synthesis; Thermo Fisher Scientific). For GFP-2A-Puro, two loxP sites (5′-ATA ACT TCG TAT AGC ATA CAT TAT ACG AAG TTA T-3′ and 5′-ATA ACT TCG TAT AGG ATA CTT TAT ACG AAC GGT A-3′) were added before and after the gene. The DNA fragments encoding expression units for aTF and GFP-2A-Puro under the control of TRE promoter were prepared via digestion with SpeI (Nippon Gene, Tokyo, Japan) and XhoI (Nippon gene), and XhoI and BstXI (Takara Bio, Kusatsu, Japan) from the custom recombinant plasmids containing chemically synthesized genes, respectively. The resultant fragments were arranged in tandem and inserted into the SpeI-BstXI-digested PiggyBac transposon vector plasmid (#PB513B-1; System Biosciences, Palo Alto, CA, USA), to generate PB/TRE_aTF/TRE_GFP-2A-Puro (Figure 1a). A DNA fragment for the scFv-Fc-2A-Puro gene optimized using human codons was chemically synthesized (GeneArt Gene Synthesis; Thermo Fisher Scientific), flanked by compatible loxP sites (5′-ATA ACT TCG TAT AGC ATA CAT TAT ACG AAG TTA T-3′ and 5′-TAC CGT TCG TAT AGG ATA CTT TAT ACG AAG TTA T-3′), and ligated into pMK-RQ (Thermo Fisher Scientific) to generate pDonor/scFv-Fc-2A-Puro (Figure 1a). To generate PB/TRE_scFv-Fc, the scFv-Fc gene fragment was obtained by digesting plasmid R2 [22] with EcoRI (Nippon gene) and MluI (Nippon gene), and inserted into EcoRI–MluI-digested PB/TRE_aTF/TRE_GFP-2A-Puro. Subsequently, a hygromycin resistance gene expression unit derived from pCEP4 (#V04450; Invitrogen) was inserted into PB/TRE_scFv-Fc to generate PB/TRE_scFv-Fc/Hyg (Figure 1a).

All plasmid digestion reactions using restriction enzymes were performed at 37 °C for 2–3 h. The constructed plasmids were transformed into *E. coli* DH5α (#9057; Takara Bio) for propagation and purified using a commercially available DNA purification kit (Qiafilter Plasmid Mid Kit, #12245; Qiagen GmbH, Hilden, Germany) in accordance with the manufacturer’s protocol. Subsequently, the endotoxin-free plasmids were analyzed to measure the concentration using a Nanodrop Spectrophotometer (Model #ND-2000C; Thermo Fisher Scientific) and stored at −20 °C until use for transfection.

### 2.3. Establishment of scFv-Fc-Producing CHO Cells

CHO cells expressing reporter genes (CHO/aTF_GFP) (Figure 1b) were established as follows. CHO-K1 cells were seeded in a 24-well plate at a density of 1.2 × 10^5^ cells/well. The next day, 640 ng PB/TRE_aTF/TRE_GFP-2A-Puro and 160 ng PiggyBac transposase expression vector plasmids (pPBase, #PB210PA-1; System Biosciences) were transiently transfected into CHO-K1 cells using 2.5 μL transfection reagent, Lipofectamine 2000 (#11668019; Invitrogen), in accordance with the manufacturer’s protocol. After 4 days of transfection, to activate the TRE promoter, the pCMV/aTF plasmid (800 ng) [18], a vector for constitutively expressing aTF, was introduced into CHO-K1 cells using Lipofectamine 2000 (2.5 μL). Subsequently, GFP-positive cells were sorted using a cell sorter (SH800; Sony, Tokyo, Japan) and selected for 4 days with 50 mg/L puromycin (#A1113803; Invitrogen). Cell cloning was performed from the gene-integrated bulk cells exhibiting strong GFP expression using the limiting dilution method, and the CHO/aTF_GFP cell line was established. GFP expression in CHO/aTF_GFP cells was suppressed after culturing for 8 days in the presence of 1 mg/L doxycycline (Dox, #D9891; Sigma-Aldrich).

Next, to replace the GFP-2A-Puro gene downstream of the TRE promoter in the reporter gene expression unit integrated into CHO/aTF_GFP cells with the scFv-Fc-2A-Puro gene, the pDonor/scFv-Fc-2A-Puro donor vector (800 ng) containing the compatible mutated loxP sites and a Cre expression vector (10 ng) [22] were co-transfected into CHO/aTF_GFP cells using Lipofectamine 2000 (2.5 μL). The GFP-extinguished cell fraction was sorted using a cell sorter, and cell cloning was performed using the limiting dilution method to establish scFv-Fc-producing CHO cells (CHO/aTF_scFv-Fc1) (Figure 1b).

To enhance the productivity of scFv-Fc, PB/TRE_scFv-Fc/Hyg and pPBase were co-transfected into CHO/aTF_scFv-Fc1 cells using Lipofectamine 2000 under similar conditions to the establishment of CHO/aTF_GFP cells. After 48 h of transfection, the cells were seeded in 60 mm tissue culture dishes (BioLite #130181; Thermo Fisher Scientific) and selected for 1 week with 4 mg/mL hygromycin (#080-07683; Fujifilm Wako). Subsequently, cell cloning was performed using the limiting dilution method to establish scFv-Fc-producing CHO cells (CHO/aTF_scFv-Fc2) (Figure 1b) with enhanced scFv-Fc expression.

### 2.4. Measurement of Transgene Copy Number

To measure the copy number of introduced genes in the recombinant CHO cells, genomic DNA was extracted from each cell line using a commercial genomic DNA extraction kit (#NPK-101, MagExtractor; Toyobo, Osaka, Japan). Quantification of the gene copy number was performed using a quantitative PCR device (AriaMx Real-time PCR System; Agilent Technologies, Santa Clara, CA, USA) via a modified version of a previously published method [22]. The primers and Taqman probes used are shown in Appendix A. Standard curves were prepared using donor plasmids (PB/TRE_aTF/TRE_GFP-2A-Puro, pDonor/scFv-Fc-2A-Puro, and PB/TRE_scFv-Fc/Hyg). The nuclear genome content of CHO cells was assumed to be 5.4 pg/cell [23] to calculate the copy number. The copy numbers of GFP and scFv-Fc genes in the cells were measured multiple times (*n* = 5) and expressed as mean values with standard deviations.

### 2.5. scFv-Fc Measurement and Metabolic Analysis

CHO cells and recombinant CHO cells were seeded at a density of 1.0 × 10^5^ cells/mL in a culture medium containing serum or 1.0 × 10^6^ cells/mL in serum-free medium, with 1 mL per well in 24-well culture plates. The cells were then cultured for 6 days. During this period, the culture media were harvested daily, and cell counting was performed using the trypan blue dye exclusion method with a cell counter (Countess, Model #AMQAF1000; Invitrogen). For the high-cell-density culture, CHO/aTF_scFv-Fc2 cells were seeded in 50 mL bioreactor tubes at a cell density of 0.5–2.0 × 10^7^ cells/mL with 10 mL of medium and cultured for 12 days. The culture medium was exchanged for fresh medium every other day, and during each change, the culture medium was harvested and cell counting was performed. The culture conditions were described in Section 2.1. All harvested medium samples were stored at −80 °C until evaluation for the following assays.

The concentration of scFv-Fc secreted in the culture medium was quantified using an ELISA method [22]. The rabbit IgG fraction of anti-human IgG F(c) (#609-4103; Rockland Immunochemicals, Philadelphia, PA, USA) and rabbit peroxidase (POD)-conjugated anti-human IgG F(c) antibodies (#609-4303; Rockland Immunochemicals) as primary and secondary antibodies, respectively, were used. Calibration curves were created using a dilution series of purified scFv-Fc or IgG (kindly provided by Dr. Naohiro Noda, Manufacturing Technology Association of Biologics (MAB), Kobe, Japan). From the scFv-Fc concentration in the medium and the number of viable cells, the specific scFv-Fc productivity (pg cell^−1^ day^−1^) was calculated. Glucose and lactate concentrations were measured using commercially available kits (Glucose Assay Kit-WST, #G264, and Lactate Assay Kit-WST, #L256; both from Dojindo, Kumamoto, Japan) with attached standards according to the manufacturer’s protocols.

Cell cultures were performed using multiple wells or tubes for each condition (*n* = 3). The measured values obtained from each sample were expressed as the mean value with standard deviation.

### 2.6. SDS-PAGE

To analyze the structure of scFv-Fc in the culture medium of high-cell-density cultured CHO/aTF_scFv-Fc2 cells, SDS-PAGE was conducted, with the culture supernatants (10 µL) being diluted 100-fold, and the samples were applied to the wells of 4–20% Mini-PROTEAN TGX precast gel (#4561094; Bio-Rad Laboratories, Hercules, CA, USA). The protein bands were stained with SimpleBlue Safe Stain (#LC6060; Invitrogen).

### 2.7. Glycan Structure Analysis

The scFv-Fc contained in the culture medium (days 8–12) of high-cell-density cultured CHO/aTF_scFv-Fc2 cells was purified using a Protein A column (rProtein A Sepharose Fast Flow, #17127901; Cytiva, Marlborough, MA, USA). The purified scFv-Fc samples were treated using a glycan composition analysis kit (EZGlyco mAb-N Kit with 2-AB, #BS-X4410; Sumitomo Bakelite, Tokyo, Japan), and glycan analysis data were obtained using the hydrophilic interaction liquid chromatography–ultra-performance liquid chromatography (HILIC-UPLC) method with a 2-AB labeled human IgG (Waters Glycan Performance Test Standard, #186006349; UVISON Technologies, London, UK) as a standard. The same samples were analyzed three times (*n* = 3), and the glycan composition distributions were expressed as the mean with standard deviation.

### 2.8. DNA Microarray Analysis

mRNA was extracted from high-cell-density cultured CHO/aTF_scFv-Fc2 cells (day 12) for transcriptomic analysis. A commercially available DNA chip for Chinese hamster (G4858A#077089 single color 8 × 60K; Agilent Technologies) was used. Microarray analysis was outsourced to Cell Innovator (Fukuoka, Japan). Enrichment analysis for functional clustering was performed using the DAVID (david.ncifcrf.gov) annotation tool. The microarray data were submitted to the ArrayExpress database at EMBL-EBI (www.ebi.ac.uk/arrayexpress (accessed on 15 November 2023)) under accession number E-MTAB-13440 (available from 1 January 2024).

## 3. Results

### 3.1. Construction of Producer CHO Cells Incorporating a High-Expression System for Target Genes with a Transcriptional Positive Feedback Loop

We have constructed expression systems in animal cells that respond to external environmental factors such as heat treatment and hypoxia. At that time, we reported the effectiveness of a high-expression system for target genes using the TetR-based transcription activation system with a positive feedback loop. In the current study, we examined the production of a model antibody using CHO cells engineered with this system. To establish CHO cells capable of high antibody production, we performed gene integration into the genome using the PiggyBac transposon vector for efficient genomic integration of the expression unit. The PiggyBac transposon vector inserts into TTAA sequences in genomic DNA, providing some degree of directionality around the transcription start site while allowing for integration throughout the entire genome [24]. Furthermore, multiple copies of the target gene flanked by inverted terminal repeats (ITRs) can be simultaneously introduced into the cell genome using a transposase enzyme [25]. We constructed a PiggyBac transposon vector plasmid incorporating both the TRE-artificial transcription factor expression unit (TRE_aTF) and the responsive expression unit containing the reporter gene (TRE_GFP-2A-Puro) on a single vector. This plasmid, along with the transposase expression vector plasmid, was introduced into CHO-K1 cells. After integrating the aTF and GFP-2A-Puro expression units under the control of the TRE promoter into the CHO cell genome, we transiently introduced a plasmid (pCMV/aTF) that constitutively expresses aTF under the control of the CMV promoter to turn the aTF expression on. After sorting GFP-positive cells and performing puromycin selection, cell clones strongly expressing GFP were obtained (CHO/aTF_GFP) (Appendix A). When CHO/aTF_GFP cells were cultured in a medium supplemented with Dox, GFP expression was completely suppressed after 8 days of culture during three to four passages (Appendix A). Furthermore, we were able to re-induce GFP expression by transiently introducing the aTF expression plasmid (Appendix A). These findings indicate that CHO/aTF_GFP cells are responsive to the transient expression of aTF, and the transgene expression can be turned “off” by the addition of Dox.

When we quantified the copy number of the introduced gene of CHO/aTF_GFP using quantitative PCR, it was determined to be 19.0 ± 0.1 copies per cell. Therefore, the strong fluorescence expression in CHO/aTF_GFP is likely due to the integration of nearly 20 foreign genes into the genome of each cell. Given the high expression of the reporter gene in CHO/aTF_GFP cells, it is expected that, by switching the reporter gene unit with the target gene, high-level expression of the target gene can be achieved. The reporter gene in CHO/aTF_GFP cells was flanked by loxP sequences, allowing it to be replaced with the target gene through a recombination reaction using Cre recombinase. Consequently, to produce scFv-Fc antibodies, we introduced the donor plasmid (pDonor/scFv-Fc-2A-Puro) containing the scFv-Fc gene, which is the gene of interest, and a selection marker gene (Puro), together with the Cre expression plasmid into CHO/aTF_GFP cells. Cells with extinguished GFP fluorescence were sorted using a cell sorter, and we obtained a CHO cell clone (CHO/aTF_scFv-Fc1) with high production of scFv-Fc. Figure 2 shows the growth curve (Figure 2a), scFv-Fc concentration (Figure 2b), and specific production rate of scFv-Fc (Figure 2c) of CHO/aTF_scFv-Fc1 cells. CHO/aTF_scFv-Fc1 cells were seeded in a 24-well plate and grown for 6 days. A decrease in growth rate was observed compared with that of the parent CHO-K1 cells, which is believed to be associated with the high expression of scFv-Fc. The scFv-Fc concentration reached 41.5 mg/L on day 6. Upon calculating the specific production rate, it averaged 18 pg cell^−1^ day^−1^ throughout the culture period.

In CHO/aTF_scFv-Fc1 cells, once the expression is turned on, it is anticipated that aTF expression is induced at a high level through a positive feedback loop under the control of the TRE promoter. Therefore, it is believed that increasing the expression units of the responsive part (TRE_scFv-Fc) would further elevate the production of scFv-Fc. Hence, we constructed a PiggyBac transposon vector plasmid (PB/TRE_scFv-Fc/Hyg) containing the TRE_scFv-Fc expression unit and a hygromycin resistance gene marker and introduced it into CHO/aTF_scFv-Fc1 cells together with the transposase expression vector plasmid. After selection with antibiotics, we established a clone with the highest scFv-Fc production (CHO/aTF_scFv-Fc2) through cloning. Measurement of scFv-Fc production in CHO/aTF_scFv-Fc2 cells revealed that, by additionally introducing the scFv-Fc gene expression unit, the production concentration reached 60.1 mg/L on day 6, which was 1.5-fold higher than for CHO/aTF_scFv-Fc1 (Figure 2b). The specific production rate was 33 pg cell^−1^ day^−1^ on average during the culture period, representing a 1.9-fold increase compared with that of CHO/aTF_ scFv-Fc1 cells (Figure 2c).

In CHO/aTF_scFv-Fc2 cells, the growth rate was further decreased (Figure 2a). When we quantified the copy number of the introduced gene in CHO/aTF_scFv-Fc cells, it was found that the scFv-Fc gene had been introduced at a copy number of 30 (16.0 ± 1.0 copies from Cre-mediated replacement, and an additional 14.0 ± 1.1 copies). The reason for the discrepancy between the copy numbers of the reporter gene and the scFv-Fc gene after Cre-mediated replacement is unclear, but it suggests the possibility of genomic rearrangements or partial deletions. From these observations, it is believed that, by increasing the number of TRE promoter-controlled scFv-Fc gene expression units, sufficient aTF was produced from the positive feedback loop, leading to improved scFv-Fc productivity.

### 3.2. scFv-Fc Production in Low-Temperature Culture

A low-temperature shift from the typical culture temperature of 37 °C to 32–33 °C enhances the production of recombinant proteins [20,21]. However, the impact of the low-temperature shift on high-expression induction by artificial transcription factors has not been investigated. Thus, the established CHO/aTF_scFv-Fc1 and CHO/aTF_scFv-Fc2 cell lines were cultured at 33 °C, and the effects on scFv-Fc production were examined. When cultured at 33 °C, minimal growth was observed throughout the culture period (Figure 2a). With regard to scFv-Fc production, CHO/aTF_scFv-Fc2 reached a concentration of 91.3 mg/L on day 6 of culture at 33 °C, representing a 1.5-fold increase compared with that upon cultivation at 37 °C (Figure 2b). The specific production rates of CHO/aTF_scFv-Fc1 and CHO/aTF_scFv-Fc2 averaged 30 pg cell^−1^ day^−1^ and 129 pg cell^−1^ day^−1^, respectively (Figure 2c), which were 1.7-fold and 3.9-fold higher than upon cultivation at 37 °C. In the case of CHO/aTF_scFv-Fc2, the specific production rate reached a maximum of 161 pg cell^−1^ day^−1^.

The measurement results of glucose and lactate concentrations in CHO/aTF_scFv-Fc1 and CHO/aTF_scFv-Fc2 cultures are shown in Figure 2d and Figure 2e, respectively. In CHO/aTF_scFv-Fc2 cultures at 33 °C, gradual glucose consumption was observed compared with that under other conditions, and lactate accumulation was also suppressed.

### 3.3. scFv-Fc Production in Serum-Free Culture

In the production of biopharmaceuticals, serum-free cultivation is preferable. Many effective serum-free media have been developed for CHO cells. To investigate whether antibody production using the Tet-transcription activation system with a positive feedback loop is effective even in serum-free medium, serum-free adaptation was conducted for CHO/aTF_scFv-Fc2. Adaptation to serum-free medium was achieved by gradually replacing serum-containing medium with serum-free medium, mixing it with serum-containing basal medium, and gradually replacing it with serum-free medium. Finally, the cells became suspended in serum-free culture.

Serum-free adapted CHO/aTF_scFv-Fc2 cells were seeded at a density of 1.0 × 10^6^ cells/mL in a 24-well plate and subjected to batch culture for 6 days at both 37 °C and 33 °C (Figure 3). When cultured at 37 °C, the cell density reached 1.3 × 10^7^ cells/mL on day 5 (Figure 3a), and the antibody concentration was finally 0.52 g/L (Figure 3b). In contrast, at 33 °C, almost no cell growth was observed (Figure 3a), but high production of scFv-Fc was observed, reaching 1.2 g/L on day 3, and finally achieving a production of 1.4 g/L. The specific production rate was 17 pg cell^−1^ day^−1^ on average for the culture at 37 °C, whereas it reached 129 pg cell^−1^ day^−1^ (maximum 186 pg cell^−1^ day^−1^) for the low-temperature culture at 33 °C (Figure 3c). Even when using serum-free medium, productivity was found to be almost equivalent to that of serum-containing medium. Glucose concentration measurements revealed that 90.8% of the initial concentration was consumed in the culture at 37 °C, while this was reduced to 68.0% for the low-temperature culture at 33 °C, involving a reduction of approximately one-quarter (Figure 3d). Lactate concentrations on day 6 of culture were 51.4 mM and 33.0 mM for 37 °C and 33 °C, respectively (Figure 3e). On the basis of these findings, the conversion rates from glucose consumption (37 °C, 38.6 mM; 33 °C, 29.0 mM) to lactate production were 0.67 and 0.57, respectively.

### 3.4. scFv-Fc Production in Semi-Continuous Culture

High-cell-density culture using suspended cells is effective in increasing the production concentration of recombinant proteins. As mentioned above, CHO/aTF_scFv-Fc2 cells adapted to serum-free suspension have been successfully created. Thus, we evaluated whether scFv-Fc could be produced at a high concentration while maintaining a high specific production rate by culturing at a low temperature in semi-continuous culture using a bioreactor tube with shaking. Culture was carried out for 12 days, with all 10 mL of the medium being replaced with a fresh one every other day. When changing the medium, if cell proliferation was observed at both 37 °C and 33 °C, the cells were reseeded to achieve the same cell density as at the initial seeding of 0.5–2.0 × 10^7^ cells/mL (Figure 4a,d). For the cultures at 37 °C, upon seeding at cell densities of 0.5 × 10^7^, 1.0 × 10^7^, and 2.0 × 10^7^ cells/mL, the average scFv-Fc concentrations over 12 days of culture were 0.76, 1.6, and 1.3 g/L (Figure 4b and Appendix A), and the specific production rates were 69, 75, and 30 pg cell^−1^ day^−1^, respectively (Figure 4c and Appendix A). The most favorable conditions at 37 °C were achieved with a seeding cell density of 1.0 × 10^7^ cells/mL, resulting in a maximum scFv-Fc concentration of 1.9 g/L (Figure 4b) and a maximum specific production rate of 92 pg cell^−1^ day^−1^ (Figure 4c).

For the cultures at 33 °C, upon seeding at cell densities of 0.5 × 10^7^, 1.0 × 10^7^, and 2.0 × 10^7^ cells/mL, the average scFv-Fc concentrations over the culture period were 1.5, 3.9, and 2.1 g/L (Figure 4e and Appendix A), and the specific production rates were 137, 189, and 51 pg cell^−1^ day^−1^, respectively (Figure 4f and Appendix A). In the culture at 33 °C, the maximum scFv-Fc concentration and specific production rate were obtained at a seeding cell density of 1.0 × 10^7^ cells/mL, reaching 5.5 g/L (Figure 4e) and 262 pg cell^−1^ day^−1^ (Figure 4f), respectively. These results indicate that, in the culture at 33 °C, the maximum scFv-Fc concentration and specific production rate were 2.9-fold and 2.8-fold higher, respectively, than in the culture at 37 °C.

Figure 5 shows glucose and lactate concentrations in serum-free high-cell-density culture. Under the 0.5 × 10^7^ and 1.0 × 10^7^ cells/mL conditions, the glucose (Figure 5a,b and Appendix A) and lactate (Figure 5d,e and Appendix A) concentrations exhibited similar profiles, with no significant differences observed. In the case of the highest glucose consumption (22.9 mM) observed at 37 °C with 2.0 × 10^7^ cells/mL, the lactate concentration was 16.1 mM, giving a conversion rate from glucose to lactate of 35%. These results suggest that, even if the medium is replaced every other day, sufficient glucose remains, indicating the possibility of reducing the frequency of medium change.

### 3.5. Analysis of Produced scFv-Fc

To investigate the structure of scFv-Fc produced in the high-cell-density culture, SDS-PAGE was performed. The results showed that, under both reducing (Figure 6a,c) and non-reducing (Figure 6b,d) conditions, bands corresponding to the target molecular weights were detected. This indicates that scFv-Fc produced at the gram-per-liter level was secreted into the medium in an intact state. The Fc region of scFv-Fc is derived from human IgG1 and is glycosylated at the N-linkage sites. Analysis of the glycan structure under low-temperature culture revealed that the glycan structural profile was highly similar to that of scFv-Fc produced at a conventional temperature (Figure 7). No abnormal glycan structures, such as non-human-type glycans, were detected.

### 3.6. Gene Expression Analysis Using DNA Microarray

Total RNA was extracted from cells cultured under high-cell-density conditions, and comprehensive gene expression analysis was performed using DNA microarrays. The results of scatter plot analysis for each culture condition are shown in Appendix A. It was found that culturing at 33 °C led to a higher number of upregulated genes globally. In fact, gene expression patterns for culture at 33 °C compared with culture at 37 °C under the same seeding cell density conditions were analyzed with a threshold of more than 2-fold or less than 0.5-fold (Figure 8a). The results showed that, for conditions of 0.5 × 10^7^, 1.0 × 10^7^, and 2.0 × 10^7^ cells/mL, 489, 644, and 505 upregulated genes were identified, while the downregulated genes numbered 8, 166, and 127, respectively. To identify distinctive genes, all genes that were upregulated in all of the seeding conditions with a Z-score threshold of 5 or higher were listed (Figure 8b and Appendix A, and Appendix A). Figure 8b shows the fold changes in the extracted genes based on the conditions with the highest production (33 °C, 1.0 × 10^7^ cells/mL). The results revealed the upregulation of genes such as the extracellular matrix matrilin 3 (Matn3) and the membrane protein SLC45A1 gene, which functions as a glucose transporter. Genes known to respond to cold shock, such as RNA-binding protein 3 (Rbm3) and cold-inducible RNA-binding protein (Cirbp) [21], were not listed. Cultures at 37 °C and 33 °C were grouped for each seeding cell density condition, and functional clustering analysis was performed for genes that exhibited a fold change in expression of >2 or <0.5 and a Z-score > 2 or <−2 (Figure 8c). The results revealed that the expression of genes related to cell adhesion and secretion significantly increased at 33 °C (Figure 8c (left) and Appendix A). Additionally, changes in the expression of lipid-related genes were observed in the downregulated genes (Figure 8c (right)).

## 4. Discussion

In recent years, the production of monoclonal antibodies in recombinant CHO cell cultures with high titers of 5–10 g/L has been reported [26]. Through the optimization of feed formulation, medium, and process parameters such as pH and temperature, antibody titers have reached up to 10 g/L over a 14 day-period, with a specific production rate of 72 pg cell^−1^ day^−1^ in a fed-batch process [26]. Improvements in cell line development and cultivation processes have significantly contributed to high-titer production [7]. In the cell line construction process, synthetic biology approaches have begun to be adopted for CHO cell engineering to achieve high-titer production of recombinant proteins [27]. In this study, we applied a gene overexpression system using an aTF based on the Tet-transcription activation system, which can strongly induce the high expression of target genes. To trigger the transient expression of aTF and induce its high-level expression, we incorporated an expression unit that forms a positive feedback loop for the transcription of aTF. We generated recombinant CHO cells in which multiple copies of an expression unit for the target gene (scFv-Fc) were integrated into the cell genome under the control of the TRE promoter driven by aTF. Furthermore, continuous production was carried out using semi-continuous culture at high cell densities of 10 million cells/mL at a low temperature. While maintaining an average specific production rate of approximately 190 pg cell^−1^ day^−1^ throughout the culture period, we achieved a production concentration of approximately 4.0 g/L of scFv-Fc antibodies. The produced scFv-Fc remained intact, and no non-human glycosylation structures were observed in it.

The PiggyBac transposon vector is a valuable tool for CHO cell line construction because it can be integrated into the CHO cell genome with multiple copies, allowing for the stable expression of a transgene [24,28]. Reports indicate that the number of copies introduced by the PiggyBac system in a single transfection is approximately 15 copies [24,25], which was also the case in this study. By using the PiggyBac transposon vector, the aTF expression unit with positive feedback of aTF is integrated with multiple copies across the entire genome, leading to the amplification of aTF expression. As a result, it was possible to induce high-level expression of the target gene controlled by the TRE promoter activated by aTF. Using GFP as a reporter gene, after screening for cells with the highest fluorescence intensity, we replaced the reporter gene (GFP-2A-Puro) with the scFv-Fc gene flanked by loxP sites using the Cre/loxP recombination system [22] to generate CHO/aTF_scFv-Fc1-producing cells. Because aTF is amplified in a positive feedback loop, it is expected that adding response units could further improve expression levels. By introducing additional copies into the cell genome using a PiggyBac transposon vector with expression units containing the TRE promoter as the response sequence to induce expression of the scFv-Fc gene, we generated CHO/aTF_scFv-Fc2 cells. In these cells, on day 6 of culture, scFv-Fc production levels were enhanced by 1.4-fold at 37 °C and 4.1-fold at 33 °C (Figure 2b). Because aTF expression levels are considered sufficient with positive feedback loop amplification, scFv-Fc expression may be further enhanced by adding more response units.

In recombinant protein production using CHO cells, serum-free media are generally employed. After suspension adaptation in serum-free media and subsequent batch culture, the scFv-Fc concentration could be increased by 10-fold while maintaining almost the same specific production rate as in serum-containing medium. The target gene expression system using the positive feedback loop amplification of aTF remained effective even when using serum-free medium, prompting further cultivation at higher seeding cell densities. The results showed that, by seeding at 1.0 × 10^7^ cells/mL and conducting low-temperature culture, scFv-Fc production was maximized. In this culture, the observed maximum lactate concentration was 3.6 mM (Figure 5e), which is less than one-tenth of the harmful lactate concentration (40 mM) often observed in high-cell-density fed-batch culture of CHO cells [29]. Consequently, under the conditions of seeding at 1.0 × 10^7^ cells/mL and low-temperature culture at 33 °C, cell proliferation was suppressed, and the cells were in a favorable state for energy metabolism, allowing for high-level production of scFv-Fc. Furthermore, in low-temperature culture, the low levels of glucose consumption and lactate accumulation suggested the potential for achieving improved production of scFv-Fc with a reduced amount of medium usage by optimizing the timing of medium replacement. Functional clustering analysis based on transcriptomic data obtained using DNA microarrays revealed the upregulation of genes related to cell adhesion and secretion of proteins for culture at 33 °C (Figure 8 and Appendix A). Specifically, significant upregulation was observed for genes associated with cell adhesion, including Ncam, Vcam, and Icam, with fold changes of 162.5, 31.0, and 4.6, respectively. It was also found that the factor Sec16, necessary for the formation of COPII-coated vesicles [30], exhibited a 59.9-fold change in expression. The reduced glucose consumption and lactate accumulation could be related to the downregulation of genes involved in lipid metabolism (Figure 8c (right)). In recent years, the development of media tailored to production cells and production formats has been actively conducted based on data obtained from transcriptomic and metabolomic analyses. By developing media suitable for the production of target proteins using aTF and optimizing the timing of medium change and feed additions, it may be possible to achieve further increases in the production of target products in the medium and a reduction in the production of inhibitory metabolic byproducts.

We previously developed gene expression systems for the high-level expression of transgenes with transcriptional amplification using an artificial gene circuit that responds to external environmental stimuli, employing the Tet-transcription activation system. To make the entire expression unit compact, we constructed an expression vector that integrates both the induction and the amplification of the target gene expression and integrated it into the cell genome [19]. In this study, we separated the switch unit, allowing for flexible selection of the trigger for the expression of the aTF. Additionally, because we employed the Tet-transcription activation system, cells incorporating this gene expression system could reset transgene expression in cultures with Dox-supplemented medium and maintain that state or induce re-expression at a later stage. In this study, we induced sustained high-level expression by turning the system “on” through transient aTF expression using pCMV/aTF. By changing the promoter acting as the trigger for aTF expression, the system can be activated at the desired timing. Transcriptomic analysis enabled us to list genes showing more than a 5-fold increase in Z-score of expression at 33 °C cultivation compared with that at 37 °C (Figure 8b). Using the regulatory regions of these genes as triggers for aTF expression, it is believed that strong gene expression can be induced in low-temperature culture. Furthermore, the potent transcription of the target gene is driven by the transcription activation domain within the aTF [31]. There have been reports of attempts to select transcription activation domains tailored to artificial transcription factors [32], structure-based development [33], and exploration of novel transcription activation domains in human cells using TAD-seq [34]. The development of aTF suitable for production using CHO cells is expected to enhance the efficiency of target substance production with this system. Recently, a translation activation technology has been reported to further improve the production of recombinant antibodies in production cell lines that have achieved gram-per-liter level production [35]. The application of this technology may lead to even higher levels of antibody expression in the high-expression system based on the aTF in this study.

In conclusion, we successfully achieved the production of scFv-Fc antibodies at the gram-per-liter level by applying an inducible gene expression system using aTF in CHO cells. In this system, the aTF expression unit was constructed as that capable of amplification through a positive feedback loop. This allowed for the massive expression of scFv-Fc from the multiple copies of the response expression unit containing the scFv-Fc gene, facilitated by a large amount of aTF with potent transcriptional activity. This system remained effective even when the serum-free medium was used, and by combining it with a temperature shift to a lower temperature, a maximum production of 5.5 g/L was achieved. The expression of the CHO/aTF_GFP cells used as the founders in this study can be reset through the addition of Dox. By selecting a trigger promoter for aTF expression, the system can easily adapt to external environmental factors such as heat treatment and hypoxia, and the activation of the expression system can be monitored by observing green fluorescence. The ability to replace the reporter genes with target genes using Cre-loxP recombination means that CHO/aTF_GFP can serve as a universal platform cell for artificial gene expression systems. The artificial gene expression system equipped with the positive feedback loop developed in this study is expected to become a new tool for constructing systems that can produce recombinant proteins at high levels.

## Figures and Tables

**Figure 1 cells-12-02638-f001:**
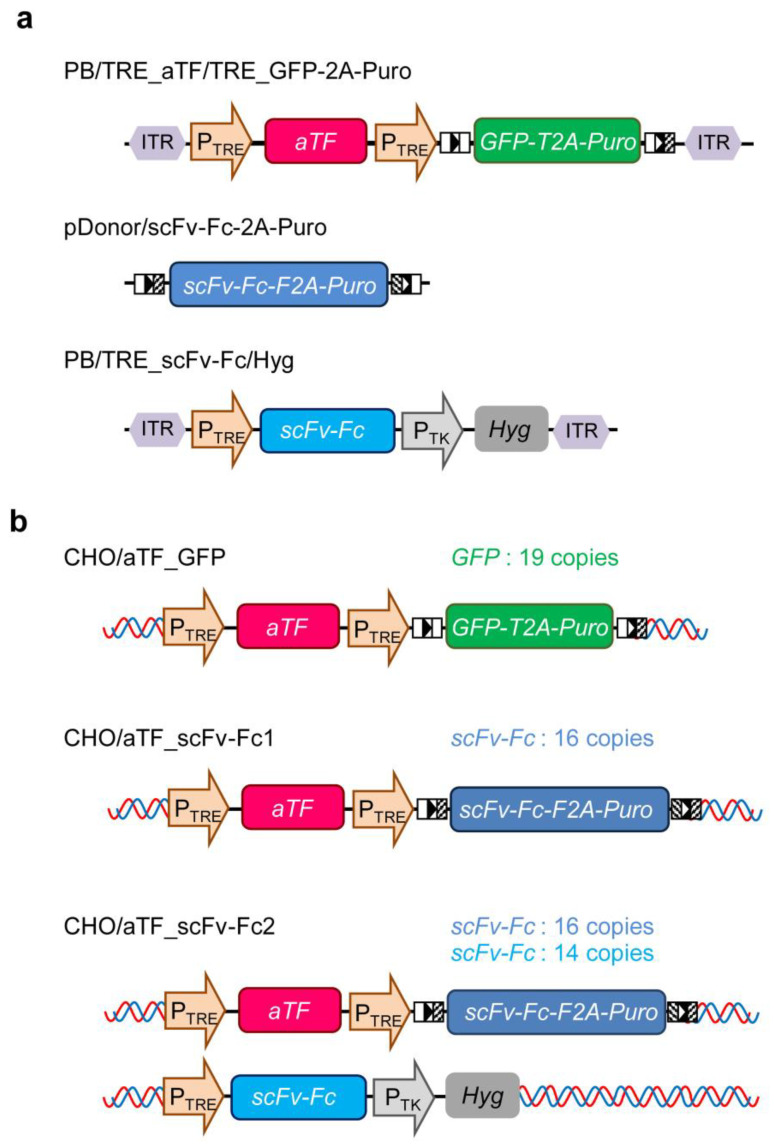
Generation of scFv-Fc-producer CHO cells with the high transgene expression system using an artificial transcription factor. (**a**) Vector constructs (PB/TRE_aTF/TRE_GFP-2A-Puro, pDonor/scFv-Fc-2A-Puro, PB/TRE_scFv-Fc/Hyg). (**b**) Recombinant CHO cells with an artificial gene expression system (CHO/aTF_GFP, CHO/aTF_scFv-Fc1, CHO/aTF_scFv-Fc2). The copy numbers of the introduced genes were measured via quantitative PCR using Taqman probes. P_TRE_, artificial synthetic promoter comprising of TRE sequence and CMV minimal promoter; aTF, tetR-VP48 fusion protein; GFP, copepod *Pontellina plumata*-derived green fluorescent protein (copGFP); T2A, 2A self-cleaving peptides derived from Thosea asigna virus; Puro, puromycin resistance gene; scFv-Fc, anti-prion single-chain antibody fused with Fc-region of human IgG1; F2A, Furin self-cleaving peptides fused with 2A self-cleaving peptides derived from porcine teschovirus-1; P_TK_, thymidine kinase promoter; ITR, inverted terminal repeat; Hyg, hygromycin resistance gene.

**Figure 2 cells-12-02638-f002:**
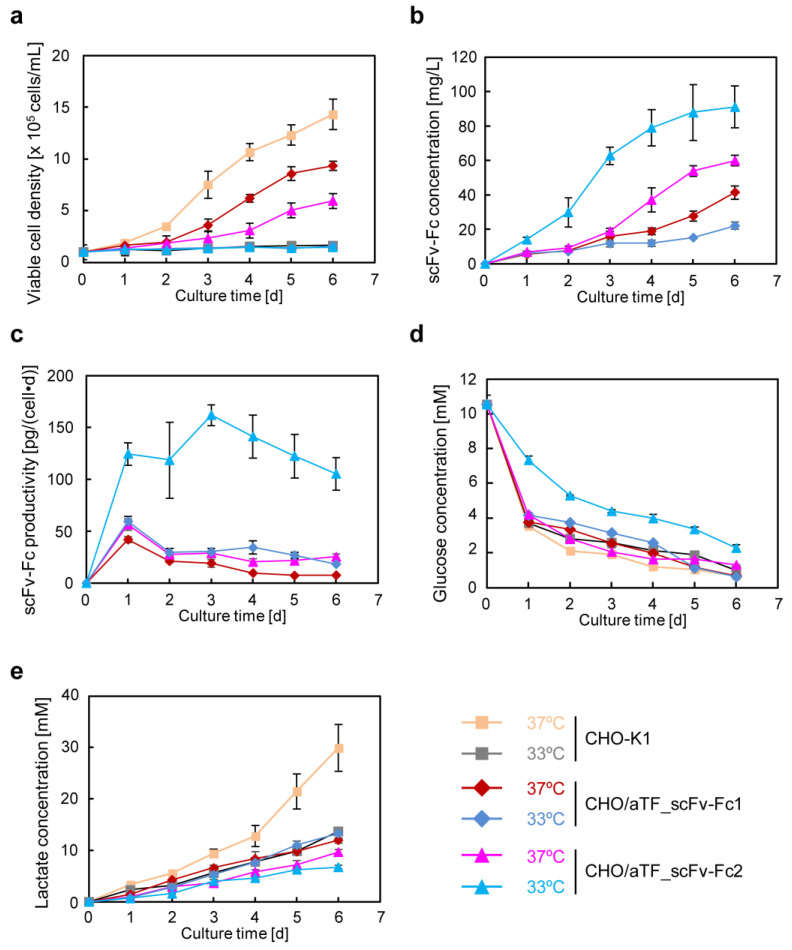
Batch culture of CHO cells with the artificial gene expression system in serum-containing medium. CHO-K1, CHO/aTF_scFv-Fc1, or CHO/aTF_scFv-Fc2 cells were seeded with 1 mL/well in a 24-well plate at a cell density of 1.0 × 10^5^ cells/mL, and cultured in F12 medium with 10% FCS for 6 days. (**a**) Cell proliferation. (**b**) scFv-Fc concentration. (**c**) scFv-Fc specific production rate. (**d**) Glucose concentration. (**e**) Lactate concentration. CHO-K1 cells (beige square, 37 °C; gray square, 33 °C); CHO/aTF_scFv-Fc1 (red diamond, 37 °C; blue diamond, 33 °C); CHO/aTF_scFv-Fc2 (pink triangle, 37 °C; light blue triangle, 33 °C). Data are expressed as mean ± SD (*n* = 3).

**Figure 3 cells-12-02638-f003:**
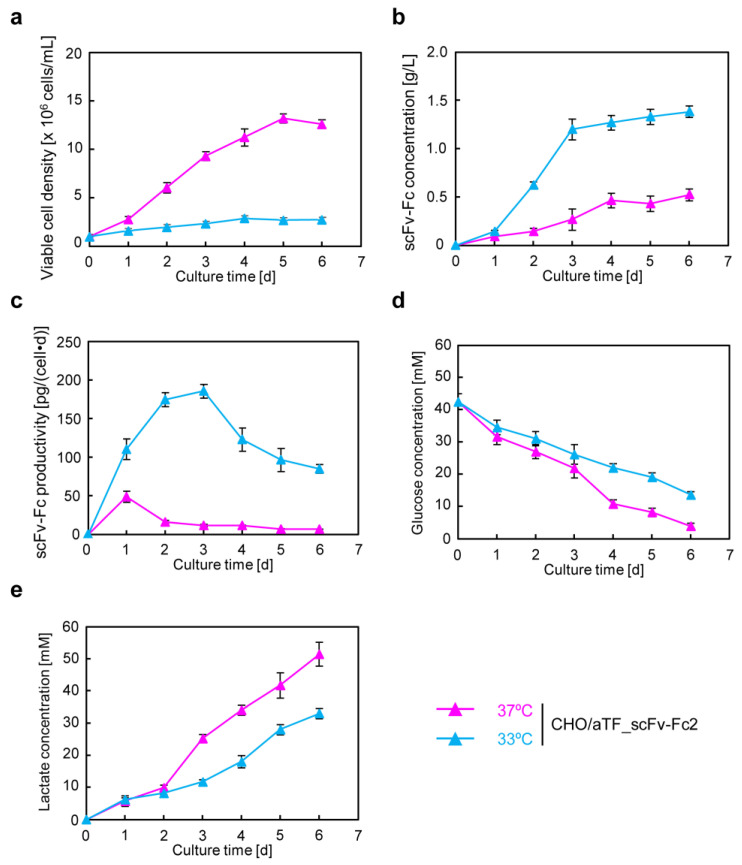
Batch culture of CHO/aTF_scFv-Fc2 cells using serum-free medium. CHO/aTF_scFv-Fc2 cells, which had been adapted and suspended in a serum-free medium, were seeded with 1 mL/well in a 24-well plate at a cell density of 1.0 × 10^6^ cells/mL, and cultured for 6 days. (**a**) Cell proliferation. (**b**) scFv-Fc concentration. (**c**) scFv-Fc specific production rate. (**d**) Glucose concentration. (**e**) Lactate concentration. Pink triangles, 37 °C; light blue triangles, 33 °C. Data are expressed as mean ± SD (*n* = 3).

**Figure 4 cells-12-02638-f004:**
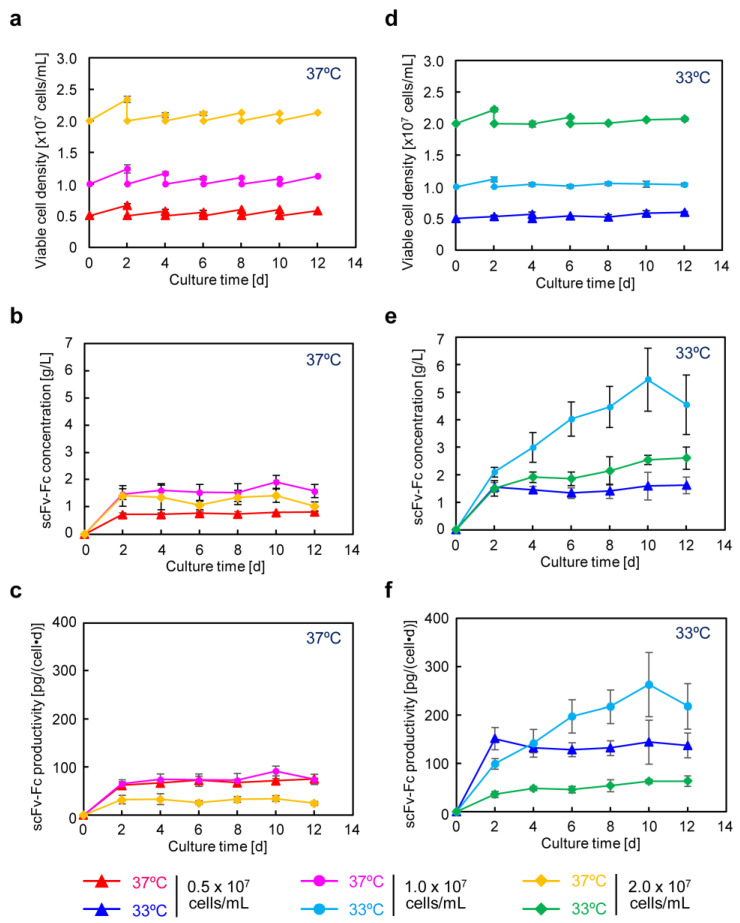
Semi-continuous culture of CHO/aTF_scFv-Fc2 cells. Cells (10 mL) were seeded in a 50 mL bioreactor tube at a cell density of 0.5 × 10^7^, 1.0 × 10^7^ or 2.0 × 10^7^ cells/mL, and cultured for 12 days. The culture medium was replaced with fresh medium every other day. (**a**,**d**) Cell proliferation. (**b**,**e**) scFv-Fc concentration. (**c**,**f**) scFv-Fc specific production rate. (**a**–**c**) 37 °C. (**d**–**f**) 33 °C. Cell seeding conditions are as follows: 0.5 × 10^7^ cells/mL (red, 37 °C; blue, 33 °C); 1.0 × 10^7^ cells/mL (pink, 37 °C; light blue, 33 °C); 2.0 × 10^7^ cells/mL (yellow, 37 °C; green, 33 °C). Data are expressed as mean ± SD (*n* = 3).

**Figure 5 cells-12-02638-f005:**
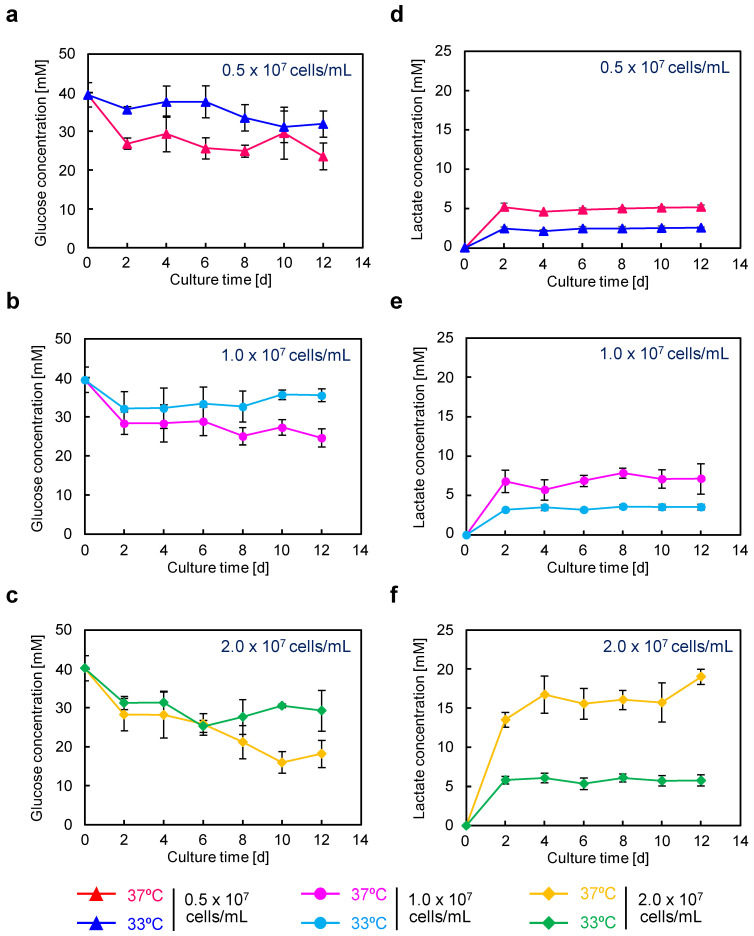
Glucose and lactate concentrations in culture supernatants of semi-continuous cultures using CHO/aTF_scFv-Fc2 cells. Glucose and lactate concentrations in the culture supernatant were measured. (**a**–**c**) Glucose concentration. (**d**–**f**) Lactate concentration. (**a**,**d**) 0.5 × 10^7^ cells/mL (red, 37 °C; blue, 33 °C), (**b**,**e**) 1.0 × 10^7^ cells/mL (pink, 37 °C; light blue, 33 °C), (**c**,**f**) 2.0 × 10^7^ cells/mL (yellow, 37 °C; green, 33 °C). Data are expressed as mean ± SD (*n* = 3).

**Figure 6 cells-12-02638-f006:**
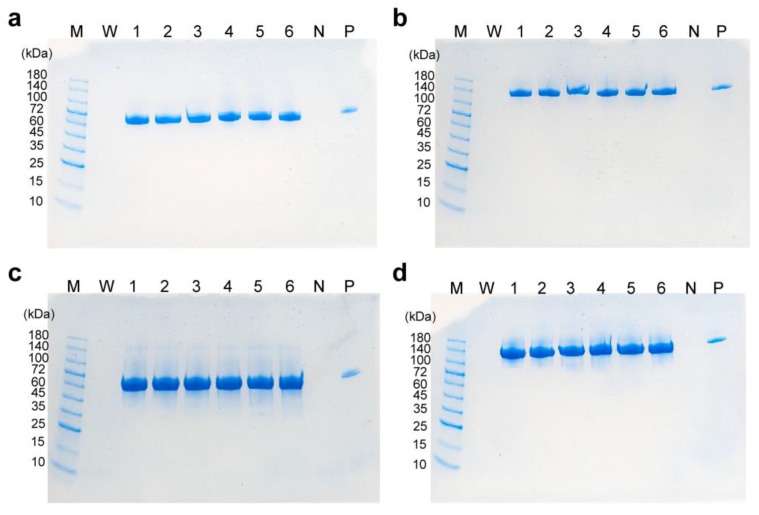
SDS-PAGE analysis of culture supernatant samples in semi-continuous culture. (**a**,**b**) 37 °C, (**c**,**d**) 33 °C. (**a**,**c**) Reducing conditions. (**b**,**d**) Non-reducing conditions. Lane M, molecular weight marker; lane W, water; lanes 1–6, culture supernatant collected on day 2 (lane 1), day 4 (lane 2), day 6 (lane 3), day 8 (lane 4), day 10 (lane 5), and day 12 (lane 6); lane N, fresh medium; lane P, purified scFv-Fc (2 µg).

**Figure 7 cells-12-02638-f007:**
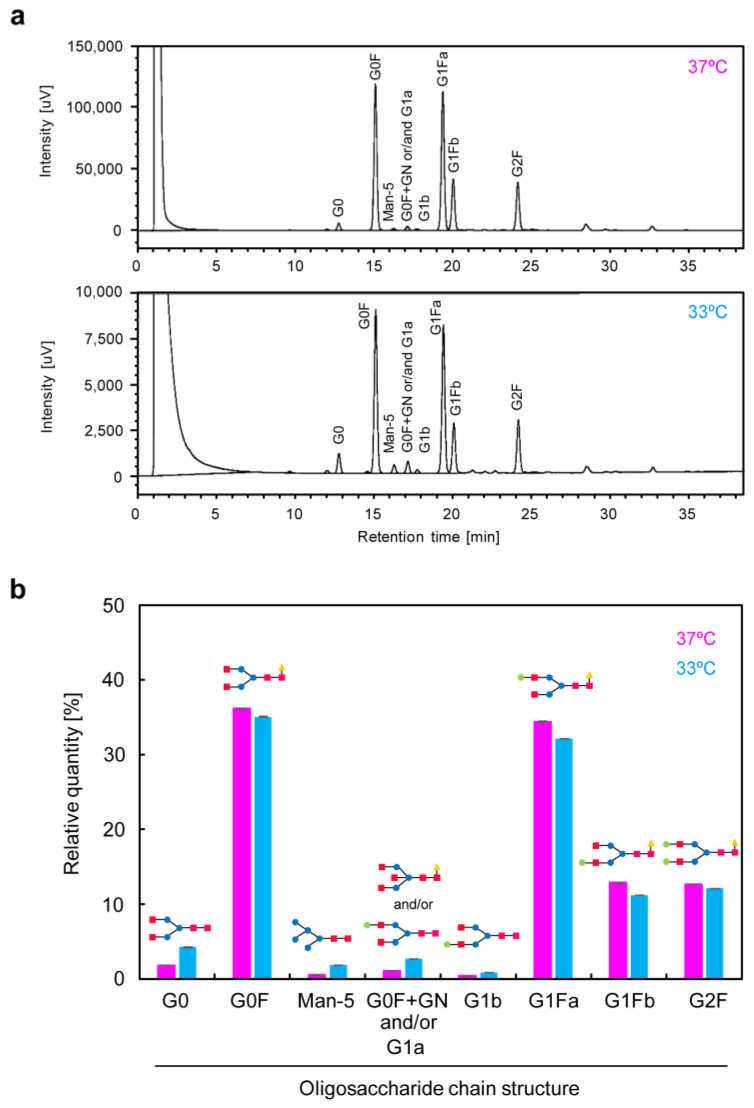
Structure analysis of *N*-linked glycans modified in scFv-Fc produced by CHO/aTF_scFv-Fc2 cells in semi-continuous culture. (**a**) HILIC-UPLC elusion profiles. Culture temperature: 37 °C (**upper**), 33 °C (**lower**). (**b**) Percentage of *N*-linked glycans detected via HILIC-UPLC analysis. Pink column, 37 °C; blue column, 33 °C. Each symbol in the schematic diagram is as follows: red squares, *N*-acetylglucosamine; blue circles, mannose; green circles, galactose; yellow triangles, fucose. Data are expressed as mean ± SD (*n* = 3).

**Figure 8 cells-12-02638-f008:**
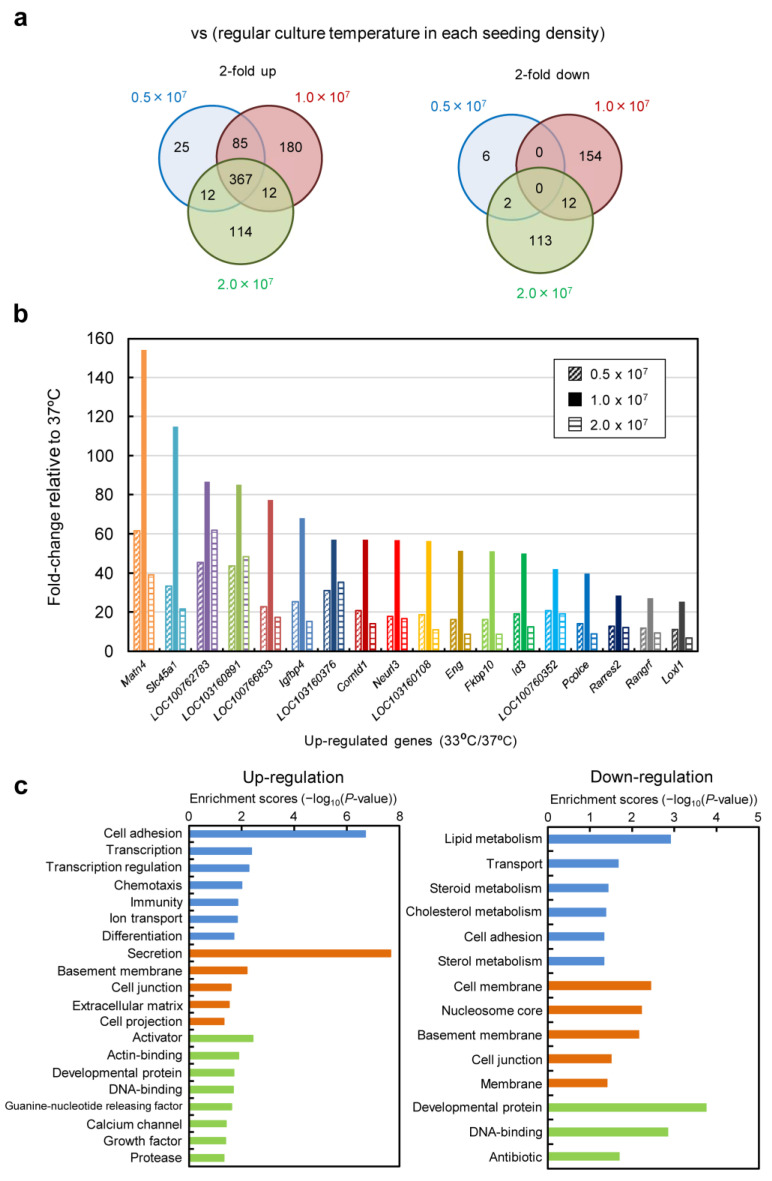
Transcriptome analysis using DNA microarray of CHO/aTF_scFv-Fc2 cells in semi-continuous culture. (**a**) Venn diagrams of the number of changes in gene expression at low temperature (33 °C) versus normal temperature (37 °C) cultured at seeding cell densities of 0.5 × 10^7^, 1.0 × 10^7^ and 2.0 × 10^7^ cells/mL. (**Left**), 2-fold up-regulation; (**right**), 2-fold down-regulation. (**b**) Genes that commonly fluctuated upward under each seeding cell density condition at low temperature. In order to narrow down the number of genes, variable genes were extracted under the condition of Z-score > 5. The rate of change in expression under the seeding condition of 1.0 × 10^7^ cells/mL that resulted in the highest antibody production is listed in descending order. The gene symbols and names are shown in Appendix A. (**c**) Functional clustering analysis. We grouped each seeding cell density condition with normal-temperature and low-temperature cultures, and extracted genes with fold change >2 or <0.5 and Z-score > 2 or <−2 via DAVID (https://david.ncifcrf.gov/ (accessed on 15 November 2023)), which was used for clustering analysis. (**Left**) Up-regulation. (**Right**) Down-regulation. Biological process (blue bars), cellular component (orange bars), molecular function (green bars).

## Data Availability

The data that support the findings of this study are available from the corresponding author upon reasonable request.

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
