# Peer review of "High-Level Production of scFv-Fc Antibody Using an Artificial Promoter System with Transcriptional Positive Feedback Loop of Transactivator in CHO Cells"

_cells, 2023, doi:10.3390/cells12222638_

Round 1
Reviewer 1 Report
Comments and Suggestions for Authors
Suggested changes but not limited to:
Please explain the method section in more detail. Examples include but not limited to:
- Line 98-99: How much media is typically used in the bioreactor tubes? Please explain the bioreaction tube design
- Plasmid construction - please elaborate on the steps, including, conditions (temperature), enzymes used (with catalog number etc.),
- Pages 135-163: Please add the following information on (1) how were the plasmids purified and how was it analyzed for concentration, endotoxin level etc., (2) amount of plasmid used in the transfection experiment, and (3) plasmid: lipofectamine ratio used in the transfection experiment
- Line 176-184: Please consider adding the incubator conditions (e.g., temperature, CO2, humidified, rpm, shaking platform)
-Line 187-194: How were the samples processed? Where the experiments performed on the same day, after harvest, or the samples were stored and analyzed later; Please consider describing ELISA method and controls used
- Line 190: Please consider adding appropriate controls
-Line 202-211: Please consider adding appropriate controls and standards
- Please consider adding the plasmid analysis and gels
Comments on the Quality of English Language
Thank you for the opportunity to review the paper.
Please do recheck the bibliography, grammatical errors, and other writing errors.
Reviewer 2 Report
Comments and Suggestions for Authors
Thank you for submitting very important work to this journal. The content of this paper is solid and there are many points of scientific interest. It is also an important paper from a practical point of view. Therefore, I think that it is possible to accept it with a little modification. Specifically, the extent of the improvement compared to the amount of antibody production reported in other papers should be more clearly described. If this point is satisfied, I would like to recommend accepting this manuscript in the journal.
Reviewer 3 Report
Comments and Suggestions for Authors
The work describes obtaining a respectable titer of an IgG antibody derivative (scFv-Fc) by applying an inducible gene expression system using an artificial transcription factor (aTF) in CHO cells.
With the glycan composition study of the cell product and the transcriptome analysis, the authors further contribute their claims.
Using standard cell line design protocol they established two stable antibody-producing cell lines with amplified antibody gene sequences.
The authors tested their cell line productivity in different cultivation systems (batch and semi-continuous), using temperature shift and serum-free media; which are standard evaluation approaches in the CHO production platform.
The research is well conducted, with a reasonable choice of controls.
Suggestion:
In the interest of understanding the figures, it is preferable to design legends in the charts/graphs, instead of having points and colors of the chart lines described in the text beneath the figures.
Round 2
Reviewer 1 Report
Comments and Suggestions for Authors
Thank you to the authors for addressing the queries.